# Implications of the Use of Biodiesel on the Longevity and Operation of Particle Filters

**Carl Justin Kamp [1,2,*] and Sujay Dilip Bagi [1]**

1.  Department of Mechanical Engineering, Massachusetts Institute of Technology, 77 Massachusetts Avenue, Cambridge, MA 02139, USA
2.  Kymanetics Inc., 45 Winthrop Shore Dr, Winthrop, MA 02152, USA
*   Correspondence: ckamp@mit.edu or ckamp@kymanetics.com

**Abstract:** While biodiesel is one of many necessary steps forward in a cleaner transportation future, alkali metal residuals, including Na and K (in the form of oxides, sulfates, hydroxides, and carbonates) originating from fuel production catalysts were found to be detrimental to emissions control components. Na + K and Ca + Mg (also biodiesel production byproducts) are regulated by ASTM-D6751 standards (American Society for Testing and Materials) to be less than 5 ppm for B100; however, the literature gives examples of physical and chemical degradation of automotive emissions catalysts and their substrates with these Na and K residuals. The purpose of this study is to investigate the impacts of ash from Na-doped biodiesel fuel (B20) on a diesel particulate filter (DPF). Investigations found that the Na-ash accumulated in the DPF has several unique properties which help to fundamentally explain some of the interactions and impacts of biodiesel on the particle filter. The biodiesel-related Na-ash was found to (1) have a significantly lower melting temperature than typical ash from inorganic lubricant additives and Ultra Low Sulfur Diesel (ULSD) fuel resulting in ash particles sintered to the DPF catalyst/substrate, (2) have a primary particle size which is about an order of magnitude larger than typical ash, (3) produce a larger amount of ash resulting in significantly thick wall ash layers and (4) penetrate the DPF substrate about $3\times$ deeper than typical ULSD and lubricant-related ash. This study utilizes numerous characterization techniques to investigate the interactions between biodiesel-related ash and a DPF, ranging from visualization to composition to thermal analysis methods. The findings suggest the need for tighter control of the thermal environment in the DPF when using biodiesel, additional/improved DPF cleaning efforts, and avoidance of unregulated biodiesel with high Na/K levels.

**Keywords:** diesel particulate filter; biodiesel ash; X-ray CT; phase transformation; sintering

## 1. Introduction

Diesel engine emissions are regulated worldwide and require complex aftertreatment systems, many of which are susceptible to chemical and mechanical degradation due to the accumulation of incombustible ash material [1]. This ash material may be derived from inorganic lubricant additives (including Ca, Zn, Mg, S, and P), engine wear, or fuel impurities (including Na/K/Ca/Mg) associated with biodiesel [2]. Wall-flow filters, including the diesel particulate filter (DPF), gasoline particulate filter (GPF), diesel oxidation catalyst on filter (DOCF), and selective catalytic reduction on filter (SCRF), are particularly sensitive to ash since it is trapped along with combustible soot particles but remains in the filter post regeneration (i.e., soot oxidation). This trapped ash increases the filter pressure-drop ($\Delta$P), decreasing vehicle fuel economy, and in some cases, can reduce the DPF full-useful lifetime (FUL). As a result, a DPF will undergo a number of cleaning events over its FUL to remove ash, to recover a portion of the ash-related pressure drop, and this typically occurs after ash accumulation levels of ~20–50 g/L corresponding to hundreds of thousands of kilometers (on-road) or thousands of hours (off-road) [3]. Filter cleaning

is performed by a number of methods, including wet/dry pressurized reverse flow and chemical soaking [4]. The majority of lubricant-derived ash accumulates within a filter as a loose powder, with adhesion mostly described by Van der Waals forces, and can be effectively removed via filter cleaning. However, some ash has increased adhesion or might have some chemical binding to the substrate and cannot be effectively removed with normal cleaning methods.

The introduction of biofuels is motivated by the need to use sustainable and renewable basestocks. These basestocks originate from multiple types of renewable sources, including vegetable oils or animal fat, and involve several production steps [5]. Biodiesel is produced via a base-catalyzed transesterification reaction of lipids (fats and oils) with alcohol (typically methanol or ethanol) in the presence of sodium hydroxide and/or potassium hydroxide catalysts to produce mono alkyl esters (biodiesel) and byproducts such as glycerol. The byproducts and starting materials must be removed to meet the ASTM standards (D7467 and D6751). The byproducts (such as glycerol) and catalysts must then be separated through a biofuel purification process, and this separation process (or impurity extraction), to some extent, controls the fuel quality due to residual sodium and/or potassium, and calcium and/or magnesium (Ca/Mg may be added in the purification step). The ASTM D6751 standard specifies that both Na + K and Ca + Mg are limited to 5 ppm for B100 (100% biodiesel) [6]. Several studies measured and/or discussed the residual Na and K found in typically available biofuels and the impacts of these impurities [7–10]. Alleman et al. [10] investigated 56 different commercially available B100 fuel samples where three samples exceeded the limit of 5 ppm Na + K. 85% of the samples tested found Na + K less than 1 ppm, and 90% found Ca + Mg below 1 ppm. Note that the preference for certain types of biofuel basestocks varies regionally; however, the process steps, including base-catalyzed transesterification and product purification, would still be relevant. It is the authors' opinion that sodium and potassium-based ash interactions with emissions control components would be a relevant issue worldwide, regardless of the specific biofuel basestock.

As described in the literature, ash from biofuels poses a number of unique issues in comparison to typical lubricant-derived ash due to the presence of alkali metal species, including sodium and potassium in the form of oxides, sulfates, hydroxides, and carbonates [11]. It has been estimated that B20 produces ash at a rate of 0.0003% by mass for 10 ppm Na + K and Ca + Mg [8]. Alkali metals in these forms may become volatilized in the presence of steam and could penetrate into the ceramic aftertreatment components and possibly be transported between components in the vaporized form [8]. In addition, alkali metals have been found to poison catalysts as well as structurally weaken some ceramics [12,13].

Regarding the impact of bio-diesel ash on engine aftertreatment components, the literature shows a variety of impacts on catalyst activity and mechanical properties, with findings ranging from no effect to significant implications. A 120K mile simulated aging with B20 (with Na and K below 1 ppm) on a light-duty application showed little to no effects of catalyst poisoning (DOC, DPF, SCR, and NOx absorption catalyst) or substrate weakening [14,15]; however, Williams et al. [16] found that heavy-duty truck catalysts are more sensitive to poisoning from biodiesel ash. In another aging study with biodiesel (B20 with Na + K and Ca + Mg at 27 times the mandated limit), a 30% decrease in DOC activity (NO oxidation) and a 5% loss in SCR catalyst activity were seen at a simulated 150K miles, while the thermal shock resistance parameter decreased by 69% at 435K miles [8]. In a study by Cavataio et al. [12], a DOC (with Pt and zeolites) was deactivated by exhaust/ash in a biofuel application. The same study found that SCR catalysts experienced a 40% reduction in NO conversion when exposed to 0.18% sodium by weight, due to vanadia and copper zeolite deactivation. Dou et al. [13] found that potassium may diffuse into a lean NOx trap substrate (cordierite), resulting in significant degradation in the thermal shock resistance. One explanation for the degradation of mechanical properties due to biofuels is given by Montanaro et al. [17], where sodium was found to react with cordierite giving sodium

aluminosilicate. Other studies [17,18] found that both the catalytic activity and mechanical strength are a function of the amount of Na + K present.

The biodiesel ash profile deposited on the walls of a DPF for a heavy-duty engine was found to be twice as thick in comparison with typical ULSD diesel fuel (at a similar amount of aging) and found a 6.8% increase in pressure drop due to ash at 150K miles [8]. This study also found that Na and K penetrated 25–30% of the filter thickness at 150K miles and penetrated the entire filter thickness at 435K miles, which increased bending strength, stiffness, and the thermal expansion coefficient while decreasing the thermal shock resistance. Ahari et al. [9] investigated multiple light-duty biofuel vehicles from a fleet where one vehicle exhibited very high tailpipe-out emissions. The vehicle was found to have significant Na and K penetration in the catalyst washcoats. In addition, the DPF in this vehicle was mechanically eroded at the filter inlet due to high Na and K levels. Ahari et al. [9] concluded that this vehicle likely had used unregulated biodiesel for approximately 100K miles and that the fuel had significant NaOH and/or KOH in the fuel, likely due to insufficient fuel separation/purification during production. The previous study points to an important point where the impurity range and fuel quality of biodiesel must be considered.

While most commercially available biodiesel can be expected to meet current impurity regulations (both Na + K and Ca + Mg are limited to 5 ppm for B100), biodiesel with higher impurities was observed to result in significant and irreversible negative impacts on engine aftertreatment components, motivating further investigations of the interactions between biofuel ash and catalytic emissions control components. The brief literature survey above shows that the impact of biodiesel-related ash on the engine aftertreatment was studied but is not well understood. The purpose of our study is to add more information to this topic by way of investigating a DPF sample that has been subjected to accelerated aging with Na-doped fuel. While this DPF sample may differ from a DPF aged with some commercial biodiesel fuels, the results presented in this study give relevant observations to the literature in order to fundamentally describe Na-based ash and its interactions with a DPF. We employ numerous analytical tools to characterize the interactions between the biodiesel ash and the DPF substrate.

## 2. Experimental Methods and Materials

This section provides a description of the ash sample generation and experimental setup for materials characterization and analysis.

### 2.1. DPF Ash Samples

The DPF ash sample was generated by a joint study between Oak Ridge National Laboratory (ORNL) and National Renewable Energy Laboratory (NREL) using a Caterpillar C9 ACERT engine (Caterpillar, Irving, TX, USA) in a dynamometer test cell [11]. The aging fuel was a B20 biodiesel doped with 14 ppm Na in the form of dioctyl sulfosuccinate sodium salt that resulted in a 14 times higher limit as specified by ASTM D6751 standard [6]. To reach the full useful life (FUL) equivalent of Na exposure, 72,500 gallons of B20 would be required that would drive for 435,000 miles with an average fuel economy of 6 mpg (miles per gallon). A SAE 15W-40 diesel engine oil (Chevron Delo LE) was used as the lubricant and was changed every 250-hour interval. Weighing of the DPF showed that 857 g of ash had accumulated at the end-of-test for ash loading (50 g/L). DPF backpressure increased continuously during the test and exceeded 6 kPa at 756 h of operation. Ash cleaning was conducted at this point, and a sample was isolated for further characterization in this study. If all of the Na in biodiesel was converted into $Na_2SO_4$ and captured by the DPF, then this would account for 738 g of ash. Lubricant oil consumption over the test was 21.2 kg which at 1 wt.% sulfated ash would contribute around 210 g. Owing to low volatility of additive species, this value is usually half of the estimated ash (around 105 g).

## 2.2. X-ray Diffraction

The X-ray diffraction (XRD) instrument utilized in this study was the PANalytical X'Pert PRO XRPD system (Malvern Panalytical Ltd., Worcestershire, UK) at the Massachusetts Institute of Technology. The X'Pert PRO has a 1.8 kW copper X-ray source with a vertical circle θ:θ goniometer and a 240 mm radius, and uses an Open Eularian Cradle (OEC) sample stage. Measurements were made at STP over 10° to 70° 2θ. Tube voltage and current were set at 40 kV and 40 mA, respectively. HighScorePlus software was used for semi-quantitative data analysis and peak fitting. Identification of compounds from the XRD pattern was made using the information on elemental data obtained from SEM-EDX (Scanning Electron Microscopy with Energy Dispersive X-ray Analysis), followed by matching peak positions of compounds available in International Centre for Diffraction Data (ICDD) crystal structure database. The weight fraction (in %) of the compounds identified may deviate from the actual weight percent in the sample as there may be amorphous material present that is not detected by XRD. Additional details on background correction and peak fitting are provided in Appendix A (Figure A1).

## 2.3. SEM and EDX

An FEI Helios NanoLab dual beam system with SEM, EDX, FIB (Focussed Ion Beam), and EBSD (Electron Backscatter Detector) was utilized for the elemental mapping and imaging of the ash agglomerates and the ash on the substrate. The NanoLab is located in the Wolfson Electron Microscopy Suite in the Department of Materials Science and Metallurgy at Cambridge University. The NanoLab instrument includes an Oxford Instruments EBSD and large area EDX detector, secondary and backscattered electron detectors, a liquid metal ion source (gallium), platinum and carbon deposition capabilities, a STEM detector, and an Omniprobe for sample manipulation.

## 2.4. Focused Ion Beam (FIB) Milling

The FIB technique utilizes an ion source originating from gallium liquid metal to mill the DPF substrate directionally along with ash and soot layers at a nanometer scale. Details of FIB operation for similar sample types and physical principles are provided in our previous study [19]. FIB milling is useful for areas under 100 μm, and cross-section ion milling (CSIM) is useful for larger distances, such as in milling through the entire thickness of a filter wall (~200–300 μm). The JEOL SM-09010 cross-section polisher (JEOL USA Inc., Peabody, MA, USA) at the Institute for Soldier Nanotechnologies (ISN) at the Massachusetts Institute of Technology was used in this study and had an argon ion beam with an accelerating voltage of 2–6 kV and a platinum alloy masking plate. Cross sections are prepared with the JEOL CSIM and then imaged with standard environmental scanning electron microscope (ESEM) and EDX methods.

## 2.5. X-ray Computed Tomography (CT)

X-ray CT is a non-destructive imaging technique that generates 3D image data, wherein a sample is placed in a rotating stage between a source of X-ray photons and the detector. This multiscale technique is useful for the characterization of emissions control systems and their interactions with lubricant-derived ash. This study utilized the Nikon/XTEK HMST 225 microCT (Nikon Metrology, Tring, UK) system at the Center for Nanoscale Systems at Harvard University. This system has a multi-metal 225 kV max X-ray source, with copper filters for low energy X-ray removal, and has a range of resolutions with voxel sizes from ~1 μm up to 127 μm, depending on sample size based on a Perkin–Elmer 12″ × 12″ detector. The X-ray CT 3D Pro software was used for reconstruction, while VGStudio Max and Avizo were used for data analysis.

## 2.6. Differential Scanning Calorimetry (DSC)

DSC requires small amounts of sample (10s of mg) and measures enthalpy change of a sample during thermal events (either endothermic or exothermic). A Q10 DSC from TA

instruments (New Castle, DE, USA) was employed, and a $N_2$ gas flow rate of 50 mL/min was used as the purge gas. Around 40 mg of sample was weighed and sealed in an aluminum pan with a max temperature range of up to 600 °C. A ramp rate of 2 °C/min was employed to generate the DSC curves in an air environment. The sample was cooled down to room temperature after the experiment and discarded.

## 3. Results

The results are further divided into subsections that present the ash and substrate characterization results from different instruments/methods along with their implications on aftertreatment system.

### 3.1. DPF Ash Composition

With vehicle mileage, the DPF traps most of the incombustible particulate matter transported in the exhaust stream, and as such, the composition of the ash in the filter is highly complex, varying with the type of engine oil, fuel composition, regeneration strategy, and intake air impurities amongst others [20–23]. Under real-world operation, a majority of DPF ash species originate from lubricant additive species (~80–90%), fuel additives including biodiesel species (~5–10%), and small amounts from engine wear, coolants and air-borne impurities such as dust [24,25]. The ash sample in this study was generated using Na-doped biodiesel (B20), thereby leading to higher quantities of Na-based compounds, their derivatives, and compounds from interactions with the filter substrate made from cordierite ($Mg_2Al_4Si_5O_{18}$). Figure 1 shows the weight fractions (in %) of various compounds identified from the XRD pattern using the search-match function assisted by EDX elemental analysis. The Na-species in biodiesel can be converted to oxides, sulfates, hydroxides or carbonates during the combustion process. Sodium sulfate ($Na_2SO_4$) originates from the Na salt in the doped fuel, while calcium sulfate ($CaSO_4$) originates from the over-based detergent chemistry in engine oil formulations [26]. The presence of the cordierite compounds can be correlated to harsher DPF cleaning protocols that could remove the washcoat and substrate along with sintered ash to the walls of the filter [23,24,27,28]. Note that some cordierite particles would be expected from XRD sample preparation (ash extraction from the filter inlet channels). Both sodium and calcium sulfate make up 62% of the ash composition, while the cordierite substrate accounts for 12%. The remaining 26% of the ash species originate from the cross-interaction of Na-ash species with the cordierite substrate as well as interaction with other ash components. For instance, sodium silicate ($Na_2Si_3O_7$) forms when Na-species from biodiesel sinters with the cordierite substrate during high-temperature exposures in the filter (>650 °C), which may occur during active regeneration or at local hot spots when soot is being oxidized [29,30]. Short bursts of temperatures in the range of 700–900 °C could significantly enhance Na ash sintering and formation of mixed-phase ash species, which may not exist initially [31]. The DPF may be exposed to > 800 °C spanning over a few cumulative hours over the full-useful life of the filter [31,32]. Similarly, a small fraction (8%) of sodium calcium sulfate ($Na_4Ca(SO_4)_3$) was also detected in the DPF ash sample, which could form by the interaction of $Na_2SO_4$ and $CaSO_4$ during high-temperature events in the DPF. Since the melting point of pure $Na_2SO_4$ is 884 °C while $CaSO_4$ is 1460 °C, the combined melting point of the system is usually lower than the individual components [33]. In line with these findings, we performed a temperature-programmed differential scanning calorimetry (DSC) measurement of the ash sample to analyze phase transitions occurring in the sample up to 600 °C.

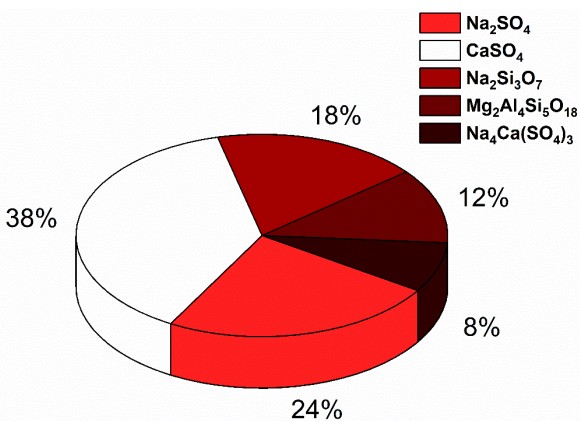

**Figure 1.** Pie chart showing the chemical makeup of ash compounds in weight fractions (%), identified using powder XRD pattern.

### 3.2. DSC (Differential Scanning Calorimeter) Analysis

DSC is a fundamental thermal analysis method used to characterize polymorphs and eutectic transitions, crystallinity, glass transition temperatures, and enthalpy of fusion, among others. Figure 2 shows the DSC trace for the Na-doped biodiesel ash sample, and the corresponding peaks are indicative of thermal events such as fusion and phase transitions. The dotted line shows the integral peak areas that are used to calculate the enthalpy change for the thermal events. Although melting points of individual components in the ash are well above 700 °C [24,25], we see the presence of exothermic peaks (negative heat flow) below 300 °C. The peak at 162.71 °C would be indicative of loss of crystallized water [34] from Ca compounds such as $CaSO_4 \cdot 2H_2O$ or other calcium sulfate semi-hydrates, while the peak at 268.32 °C would be a tell-tale sign for the onset of sintering. As a general rule of thumb, the minimum melting temperature for a mixture of compounds occurs at a certain composition of components, which is called the eutectic point. However, the melting point depression (lowering of the melting point of a pure component) occurs in the presence of other impurities or compounds with lower melting points. This is energetically favorable as the melting of an impure solid into an impure liquid has a larger change in entropy than melting a pure solid into a pure liquid; this large change in entropy corresponds to a lower melting temperature. The peak at 467.21 °C can also be explained using melting point depression. The practical implication of a lower melting point of biodiesel ash would result in increased ash sintering, possible higher pressure drop across the filter, possible fouling of active sites, and the need for using more effective cleaning techniques to service the filter.

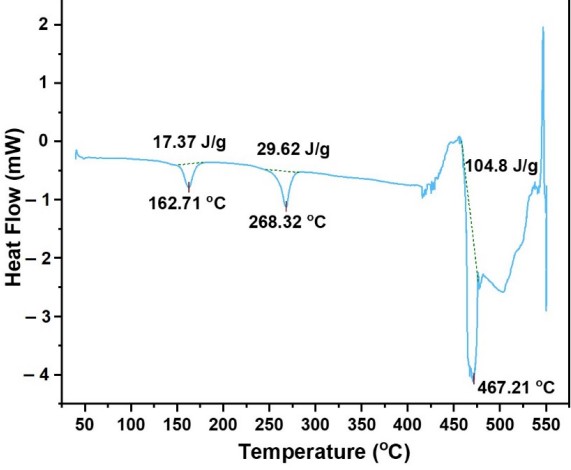

**Figure 2.** DSC trace of the biodiesel ash sample showing endo- and exothermic transformations along with the corresponding enthalpy of fusion (in J/g).

*3.3. Na-ash Imaging and Elemental Analysis*

SEM imaging of the ash from this sample showed characteristics deviating from the norm whereby a bimodal primary ash particle size was observed with one type of particle in the ~500 nm to 2 μm range and the other being ~5 μm to 10 μm range. The larger particles are found to be sodium sulfate ($NaSO_4$) particles and are shown in the SEM images in Figure 3 (top and bottom left) and confirmed by EDX mapping. The EDX mapping shows a clear separation between Na and Ca/Zn, indicating that the large particles are Na-based and the small particles are Ca/Zn-based. This separation is also seen in the mapping of sulfur and phosphorus, whereby the sulfur is found in $CaSO_4$ and $NaSO_4$, while the phosphorus is found in $Zn_3(PO_4)_2$ or $Zn_2P_2O_7$.

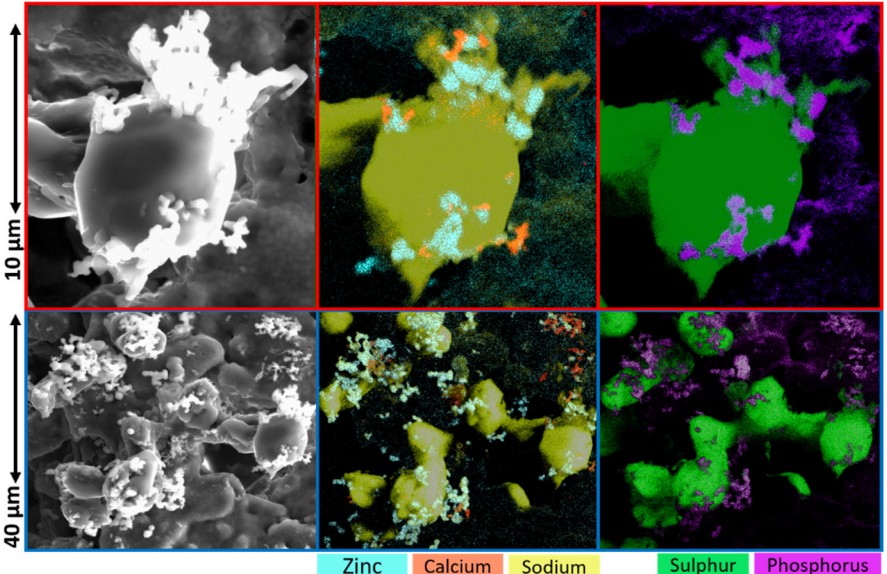

**Figure 3.** SEM imaging (**left**) and EDX mapping (**center** and **right**) of biofuel-related ash particles showing a distinct separation between Ca/Zn ash and Na ash.

The SEM images in Figure 3 show that the Na-based ash particles are significantly larger and are approximately the same size as substrate surface pores. Rather than typical small ash particles (ash particles which are ~200 nm–2 μm or ~10–20 times smaller than the surface pore size) slowly filling substrate surface pores, the Na-based ash is shown in the figure to quickly fill surface pores as shown in Figure 4 (center and right). Since particle trapping in surface pores can contribute over 50% of the overall filter pressure drop in some cases [35], the fact that Na-ash behaves so differently than typical Ca/Zn/Mg ash likely has a large impact on pressure drop. The bimodal distribution of ash particle size likely has an effect on the porosity of ash in pores as well; thus, both the pore-filling mechanism and pore-filled properties likely deviate from the norm and require further study to fully understand. Furthermore, SEM images show that the larger particles appear to sinter/melt into the ceramic (cordierite) filter surface while the smaller particles remain separate from the ceramic surface. Figure 4 (left) shows that cracks are observed in the substrate surface near an area where Na-ash has bonded to the surface.

The SEM and EDX mapping images show that sodium-based ash particles tend to be around one order of magnitude larger than typical ash (Ca/Zn/Mg-based) and that they melt/sinter to the cordierite substrate surface. The preferential melting/sintering of Na-ash can be explained by the lower melting temperature in comparison to typical ash. This point will be explored further in the discussion section.

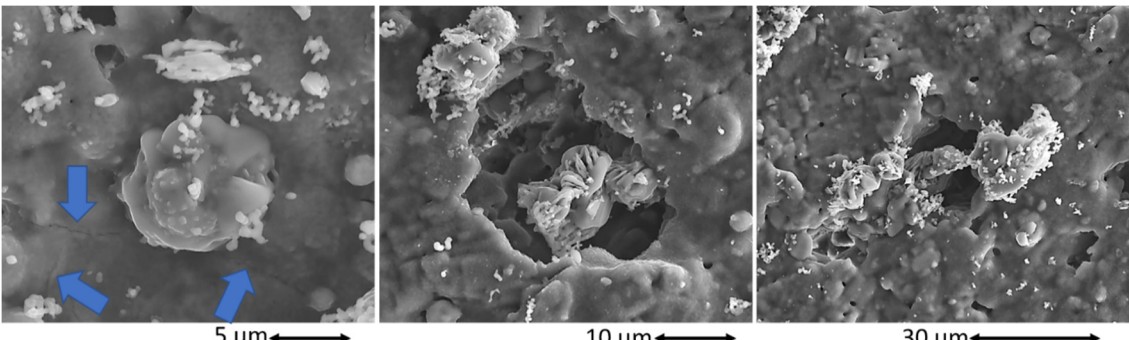

**Figure 4.** SEM images showing substrate cracks (near arrows) where Na-ash binds to the surface (**left**) and individual surface pores with large Na-ash particles (**center** and **right**).

### 3.4. Ash Agglomeration Profiles

X-ray computed tomography was utilized to observe the ash agglomeration on the channel scale, where Figure 5 shows the axial ash profile at low resolution (Figure 5a) and at higher resolution (Figure 5d), as well as the radial ash profile upstream (Figure 5b) and downstream (Figure 5c). The CT data (especially in Figure 5b) clearly shows that the wall ash layer in this sample is significantly thick, around 200 μm, in many locations upstream from the ash plug. In general, this wall thickness is far greater than that seen in typical ash-loaded filters. A previous study [8] observed increased filter pressure drop with biodiesel ash, which could be partially explained by Figure 5b. Figure 5c shows that the wall ash thickness downstream of the filter (upstream from the ash plug but in the outlet section of the filter) is extremely thick, over 500 μm in some areas, which is not often observed in filters with typical ash. Note that the average upstream wall ash thickness for this sample was found to be 180 μm by the study which generated this sample [11].

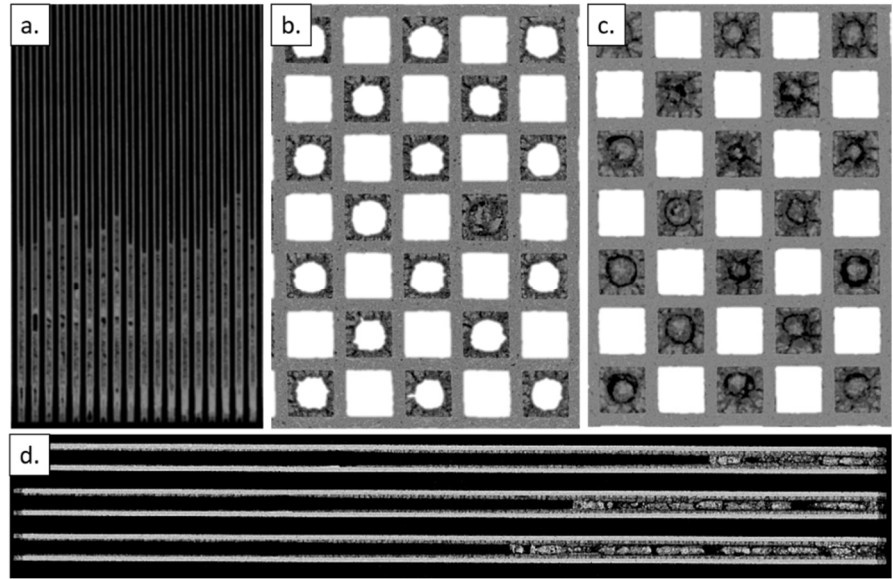

**Figure 5.** X-ray CT images of the accelerated aging sample, where (**a**) shows a low-resolution axial cross-section, (**b**) shows the high-resolution radial cross-section upstream section with a thick wall ash layer, (**c**) shows the downstream section with ash plugs, and (**d**) shows the high-resolution axial cross-section.

The authors of this study hypothesize that the thicker wall ash layers may be related to the large particle sizes of the sodium ash. Furthermore, the large particle sizes of Na-ash could influence ash transport and will be discussed in a future study. Under normal circumstances, the significantly thick all ash layers would increase the filter pressure drop

to a level that may trigger frequent filter regenerations. It has been seen in the literature that the increased pressure drop due to biodiesel-related Na-ash is observed but may only be 10–20% higher than ash without biofuel-related species in some studies and several times larger in others. This may be influenced once again by the Na-ash particle size, which likely has an impact on the porosity/permeability of the wall ash layer and will be discussed in a subsequent article.

### 3.5. Na-Ash Penetration

Cross-section ion milling by argon ions (via the JEOL Cross Section Polisher) allows for the direct observation of subsurface details, which, in this case, is the penetration of ash particles into the pore networks within the ceramic filter substrate. Single filter channels were carefully separated and cross-sectioned, followed by SEM imaging and EDX line scans to determine sodium penetration. This method was utilized in a previous study [19] to observe finer details of sub-surface ash. Figure 6 shows an SEM image of the channel cross section (bottom) and the EDX line scan plot (top).

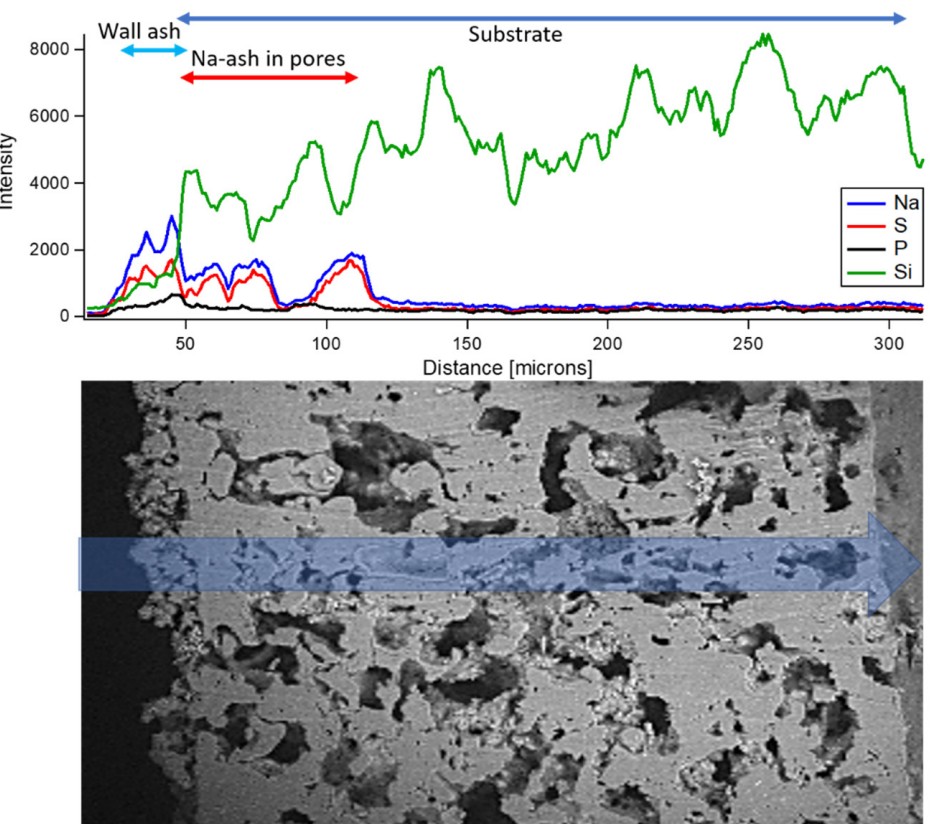

**Figure 6.** Cross-section ion milling showing Na-ash depth penetration of approximately 75 μm (**top**) and SEM image of cross-section (**below**).

Figure 6 shows that the wall ash layer is around 50 μm thick in this specific location which was taken from the upstream section of the filter, while the ash penetration is approximately 75 μm. Typical ash penetration is around one pore diameter, or 10–25 μm in DPFs for ULSD fuel engines; thus, the Na-ash penetration in this sample appears to be at least three times as deep. Recalling the findings in a study by Sappok et al. [35], ash penetration into the substrate accounted for around 6% of the total mass of ash in the filter yet resulted in up to 50% of the total pressure drop. The observations in this study show significantly greater ash penetration, implying additional ash material in the substrate porous network compared to typical lubricant-derived ash, which may have a large impact on ash-related pressure drop.

## 4. Discussion

The data collected in this study shows that presence of Na in lubricant-derived ash results in a number of unique characteristics which deviate from typical ash (ash from inorganic lubricant additives). All findings in this study are related to three concepts, whereby (1) the sodium and potassium-based fuel production catalyst residuals leftover in the biodiesel fuel led to additional ash in the DPF, (2) the Na/K-based ash has a lower melting temperature in comparison to Ca/Zn/Mg-based ash, and (3) Na-based ash was found to be semi-spherical particles which are about an order of magnitude larger than typical ash particles. This first point was summarized by Lance et al. [11] in the study that generated this sample, where the simulated biodiesel aging increased the total ash amount by a factor of approximately two, which also negatively affected the filter pressure drop. Note that the Na-ash in this DPF sample is generated from Na doped in the B20 biodiesel fuel but that the fuel doping and DPF aging were equivalent to a full useful life of the filter using B20 meeting the ASTM standard (Na + K < 5 ppm). Frequent regenerations were necessary during the filter aging protocol due to a continuously increasing pressure drop which grew to over 6 kPa [11]. The X-ray CT imaging of the ash accumulation in the present study confirmed the presence of additional ash and thick wall ash layers (up to 500 μm thickness in some areas). There is a clear relationship between the greater amount of ash due to biodiesel and the increased filter pressure drop, as discussed by Lance et al. [11] and Williams et al. [8]. The practical implication of this point is the need for increased filter cleaning efforts (more frequent cleaning/service) since the filter will be filled with ash faster in comparison to ULSD fuel. Relevant DPF substrate design characteristics which may be important in accommodating biodiesel ash include channel symmetry, porosity, pore size distribution, pore shape, material (i.e., cordierite, silicon carbide, or in some cases, aluminum titanate), cells density (i.e., cells per unit area), and washcoat thickness. Asymmetrical filter channels where the inlet channels are larger than the outlet channels may aid in the excess ash accumulation with biodiesel applications.

The second major finding is that biodiesel-related alkali metal ash (Na/K as sulfates, phosphates, oxides, and hydroxides) has lower melting temperatures than typical ash (Ca/Zn/Mg-based). The relevant bulk melting temperatures are as follows: $T_m(CaSO_4)$ = 1460 °C, $T_m(MgSO_4)$ = 1124 °C, $T_m(Zn_2P_2O_7)$ = 1010 °C, $T_m(Na_4Ca(SO_4)_3)$ = 953 °C [36], $T_m(NaSO_4)$ = 884 °C. Note that the effective melting temperatures of these ash species are lower than those of the listed bulk species melting temperatures due to the melting point depression concept, explained by the high surface-to-area-to-volume ratio for nanometer and micron-sized particles. A study by Aravelli et al. [31] found that the DPF under standard operation may see temperatures as high as 1000 °C for short periods of time and that temperatures around 900 °C may be common for a cumulatively short amount of time. The authors of this study found, by way of in situ XRD described in a previous study [37], that the irreversible chemical changes (i.e., oxidation, melting, sintering) can occur rapidly in the range of 5–30 seconds at temperatures over 800 °C. Typical ash, comprised of phosphates and sulfates of Ca/Zn/Mg, tend to be stable up to 750–800 °C. However, the lower melting temperature species such as $NaSO_4$ and $Na_4Ca(SO_4)_3$ are observed to melt and sinter from the 200 °C to 600 °C high-temperature spikes of the aging protocol in the generation of the samples investigated in this study [11]. Note that the 600 °C thermal spikes happened during filter regeneration, and the exothermic oxidation of soot would likely cause the filter temperature to be slightly higher than 600 °C but likely below the 750–800 °C stability limit for typical ash. DSC measurements in this study found a peak at 467.21 °C, suggesting the onset of ash melting at significantly lower temperatures due to the presence of Na-ash. Due to the lower melting temperatures of the sodium ash species, ash-substrate hybrid species were observed with the XRD, including $Na_2Si_3O_7$, indicating the chemical binding of ash to the substrate. When ash binds chemically to the substrate, there are several associated negative effects, including difficulty/inability in filter cleaning, embrittlement of the substrate, pore blocking, and catalyst masking. A further discussion of this topic will be included in a future study whereby the lower ash

melting temperature with the addition of biofuel species will be evaluated in terms of the eutectic mixture. The practical implication from this point is that the thermal conditions of the filter will likely need to be better controlled to avoid temperatures over ~600–650 °C and to investigate cleaning techniques, which might be able to remove any of the sintered material. Detailed investigations of Na and K-based ash on different DPF washcoats (varying washcoat composition and thickness) and substrate materials (cordierite, silicon carbide, and aluminum titanate) in future studies may highlight material combinations which are less prone to Na/K ash sintering and related durability issues.

Biodiesel-related species contribute to additional accumulated ash material as shown in the literature, and the findings in this study show that the Na-ash exhibits unique properties, including large individual particles, which are about an order of magnitude larger than typical ash particles, and the ability to penetrate deeper into the substrate. The larger particles likely have an impact on ash transport within the filter, ash porosity/permeability, and ash-pore interactions. A previous study [24] discusses ash transport as being either regeneration-induced or flow-induced; however, ash particle size plays a large role in both of these mechanisms. For example, larger primary ash particles are more likely to be sheared from the surface of the inlet channel and become re-entrained in the flow and pushed towards the back, thus possibly increasing flow-induced transport. The regeneration-induced transport of the larger $NaSO_4$ particles may be decreased due to the lower contact area (surface area) of the particles and adhesion to soot and may also be inhibited due to the melting/sintering from the low melting temperature of Na-species. In addition to ash transport, the large particle size likely has some effect on how ash interacts with the substrate pores. SEM images in Figure 4 show that filter surface pores appear to be filled by a small number of large $NaSO_4$ particles. Thus, the pore filling, i.e., the ash depth penetration regime that is seen as the first steep increase in filter pressure drop due to ash loading, likely happens by way of an accelerated and possibly different mechanism in comparison to typical Ca/Zn/Mg ash. In addition, the packing of ash particles within pores, and thus the ash porosity/permeability, is likely different for the case of Na-based ash, which has a strong bimodal distribution compared to a more unimodal distribution of typical ash. This point, however, has not been studied in detail in the current work and will be discussed in a future study. Most likely, the substrate pore size (especially the pore opening size on the inlet channel face) will be an important design factor for DPFs used in biodiesel applications. For example, a larger pore size (i.e., >25 μm) may allow the large Na-based ash particles to penetrate into the substrate's porous network, while smaller pores may effectively block these particles, potentially leading to a large effect on the depth penetration-associated filter pressure drop.

Ion milling was utilized in this study to investigate the depth of penetration of sodium-based ash, where it was found to penetrate around three times deeper into the substrate than typical ash. As described in the introduction section, several previous studies discuss Na/K-based ash as being more volatile, with the possibility of transitioning into a vapor form to pass through the substrate wall and even from one aftertreatment component to the next component downstream. While the current study only investigated the DPF and did not look at other downstream components (i.e., SCR), it found significant penetration of Na-ash into the filter substrate wall thickness. This finding has numerous implications, including substrate embrittlement, possible catalyst poisoning and fouling, flow restrictions, loss of structural porosity, and reduced active surface area. Note that the study by Lance et al. [11] found that the sodium slightly increased the structural integrity of the DPF, but that there might be a new lower-strength failure mechanism, while other studies have found that sodium and/or potassium weaken or degrade the DPF structure. Perhaps additional cleaning investigations could determine if the Na-ash penetration is reversible or at least controllable.

## 5. Conclusions

This study utilized a multi-instrument approach to investigate the characteristics and impacts of lubricant- and fuel-derived ash generated in an accelerated aging experiment with biodiesel. A DPF aged with a fuel-doped 14 ppm sodium concentration in order to represent a full useful lifetime-aged emissions control system within the ASTM standard (5 ppm Na + K) was examined with XRD, SEM + EDX, ion milling, DSC, and X-ray CT in order to fundamentally observe unique qualities of Na-based ash. The ash investigated in this study, as well as the interactions between ash-substrate, are found to deviate from typical lubricant-derived ash (Ca/Zn/Mg-based) in several notable ways and are summarized below.

1.  The ash in this study was found to have a bimodal particle size distribution, where the Ca/Zn/Mg-ash primary particle size was in the 0.5–2 $\mu$m range, while Na-ash was around 10 $\mu$m and larger. This finding was observed with SEM imaging and validated by EDX mapping. These ash particles were sufficiently large enough to fill filter surface pores with a small number of particles.
2.  The large $NaSO_4$ ash particles were observed to melt and sinter to the substrate surface, which is explained by the lower melting temperature of $NaSO_4$ (approximately 600 °C lower melting temperature compared to $CaSO_4$). XRD data showed the existence of $Na_2Si_3O_7$, indicating the chemical binding of sodium ash to the cordierite substrate surface. DSC measurements show a peak at 467.21 °C, which suggests the onset of ash melting at a temperature significantly lower than ash primarily from inorganic lubricant additives.
3.  The doped biofuel was found to produce additional ash (compared to ULSD), resulting in significantly thick wall ash layers (up to 500 $\mu$m in some areas), as shown by X-ray CT axial and radial internal cross sections.
4.  Ion milling enabled filter cross-sectioning, showing that Na-ash penetrates over 75 $\mu$m into the filter wall thickness, which is approximately three times deeper than typical ULSD ash.

The implications of the findings based on the fuel-doped Na ash from an accelerated aging system listed above relate to thermal regulation of the aftertreatment system and filter cleaning methods and service intervals. In general, biodiesel produces additional ash in comparison to ULSD and necessitates several considerations. First, the additional ash means that a filter will accumulate ash faster and that the filter pressure drop will likely increase accordingly. Thus, a higher filter service/cleaning interval is likely needed for biodiesel applications. In addition, the cleaning methods will likely need to be designed specifically to deal with the Na/K-ash, which (1) interacts more with the substrate (deeper penetration), (2) has lower melting temperatures, and (3) has much larger particles. Second, since the Na/K-ash has significantly lower melting temperatures, tighter control of the thermal environment in the filter is suggested. In general, ash sintering/melting appears to be irreversible, meaning that a filter can be seriously damaged by a short, high thermal excursion. Thus, it becomes increasingly important to design the filter system to avoid high temperatures. The findings in this paper also support the idea that unregulated biodiesel, or rather biodiesel with Na + K exceeding the mandated ASTM standard of 5 ppm, can easily destroy the complex and expensive aftertreatment components required to meet emissions regulations. While the increased use of biodiesel is, for most reasons, an important step forward in cleaner transportation technology, there are several emissions system design and operational considerations needed to accommodate the biodiesel-related ash. Looking forward, it will be important to compare the fundamental observations of Na-based ash discussed in this paper with real-world aged emissions systems components.

**Author Contributions:** Conceptualization, C.J.K. and S.D.B.; methodology, C.J.K. and S.D.B.; software, C.J.K. and S.D.B.; validation, C.J.K. and S.D.B.; formal analysis, C.J.K. and S.D.B.; investigation, C.J.K. and S.D.B.; resources, C.J.K. and S.D.B.; data curation, C.J.K. and S.D.B.; writing—original draft preparation, C.J.K.; writing—review and editing, C.J.K. and S.D.B.; visualization, C.J.K. and S.D.B.;

supervision, C.J.K.; project administration, C.J.K.; All authors have read and agreed to the published version of the manuscript.

**Funding:** This work was supported in part by the MIT Consortium to Optimize Lubricant and Diesel Engines for Robust Emission Aftertreatment Systems, which ended in 2019. No other funding contributed to this study.

**Data Availability Statement:** Data from this study is available upon request from the corresponding author.

**Acknowledgments:** The authors would like to thank Michael Lance, Todd Toops, and their team at Oak Ridge National Laboratory for providing the artificially aged DPF sample. The authors would also like to thank Charles Settens and Suyong Han at MIT for fruitful discussions on XRD data and peak refinements. The SEM and EDX analysis was performed at the Wolfson Electron Microscopy Suite in the Department of Materials Science and Metallurgy at the University of Cambridge, and the authors thank John Walmsley and Simon Griggs for guidance. The ion milling was performed at the Institute for Soldier Nanotechnologies at the Massachusetts Institute of Technology, and the authors would like to thank Bill DiNatale for his help. X-ray CT work was performed at the Center for Nanoscale Systems at Harvard University, and the authors would like to thank Greg Lin for his guidance.

**Conflicts of Interest:** The authors declare no competing financial interest.

## Abbreviations

| | |
|---|---|
| ASTM | American Society for Testing and Materials |
| CSIM | cross-section ion milling |
| CT | X-ray computed tomography |
| DOCF | diesel oxidation catalyst on filter |
| DPF | diesel particulate filter |
| DSC | differential scanning calorimeter |
| EBSD | electron backscatter detector |
| EDX | energy dispersive X-ray spectroscopy |
| FIB | focused ion beam milling |
| FUL | full useful life |
| GPF | gasoline particulate filter |
| ICDD | International Centre for Diffraction Data |
| ISN | Institute for Soldier Nanotechnologies |
| MPG | miles per gallon |
| NOx | nitrous oxides |
| OEC | open Eularian cradle |
| PDF | powder diffraction file |
| SAE | Society of Automotive Engineers |
| SCR | selective catalytic reduction |
| SCRF | selective catalytic reduction on filter |
| SEM | scanning electron microscopy |
| STEM | scanning transmission electron microscopy |
| ULSD | ultra-low sulfur diesel |
| XRD | X-ray diffraction |
| XRPD | X-ray powder diffraction |

## Appendix A

*Appendix A.1 Phase Identification with HighScore Plus Using XRD Pattern*

The raw diffraction file generated from the XRD instrument is inserted into the HighScore Plus software and subject to background correction based on an iterative method [38–40]. The possible peaks which are above the noise threshold are chosen, which is helpful in matching the reference patterns to model compounds. The "search-match" function is used to specify the presence of elements such as Na, S, P, Zn, O, and Ca, using one of the restrictions, such as "All of", "At least one of", and "None of". The resulting

patterns are ranked based on their scores, and the most probable compounds with a 'Star (S,+)' pattern are chosen to fit the XRD pattern from the sample. The chosen patterns can be fit using the default profile fit method to obtain semi-quantitative analysis of the compounds present in the sample. Note that in Figure 3, the EDX maps show the presence of Zn- and P-based compounds that originate from lubricant anti-wear additives such as ZDDP (Zinc dialkyl dithiophosphates). However, the Zn- and P-based compounds were not identified in the XRD pattern. This discrepancy could have been due to: (1) a smaller quantity of ZDDP compounds resulting in a lower S/N ratio in the pattern, (2) overlapping peaks with other ash components, or (3) semi-crystalline or amorphous phase of Zn- and P-based compounds. The PDF (Powder Diffraction File) cards used for the fitting along with the peak list and raw files can be made available on request.

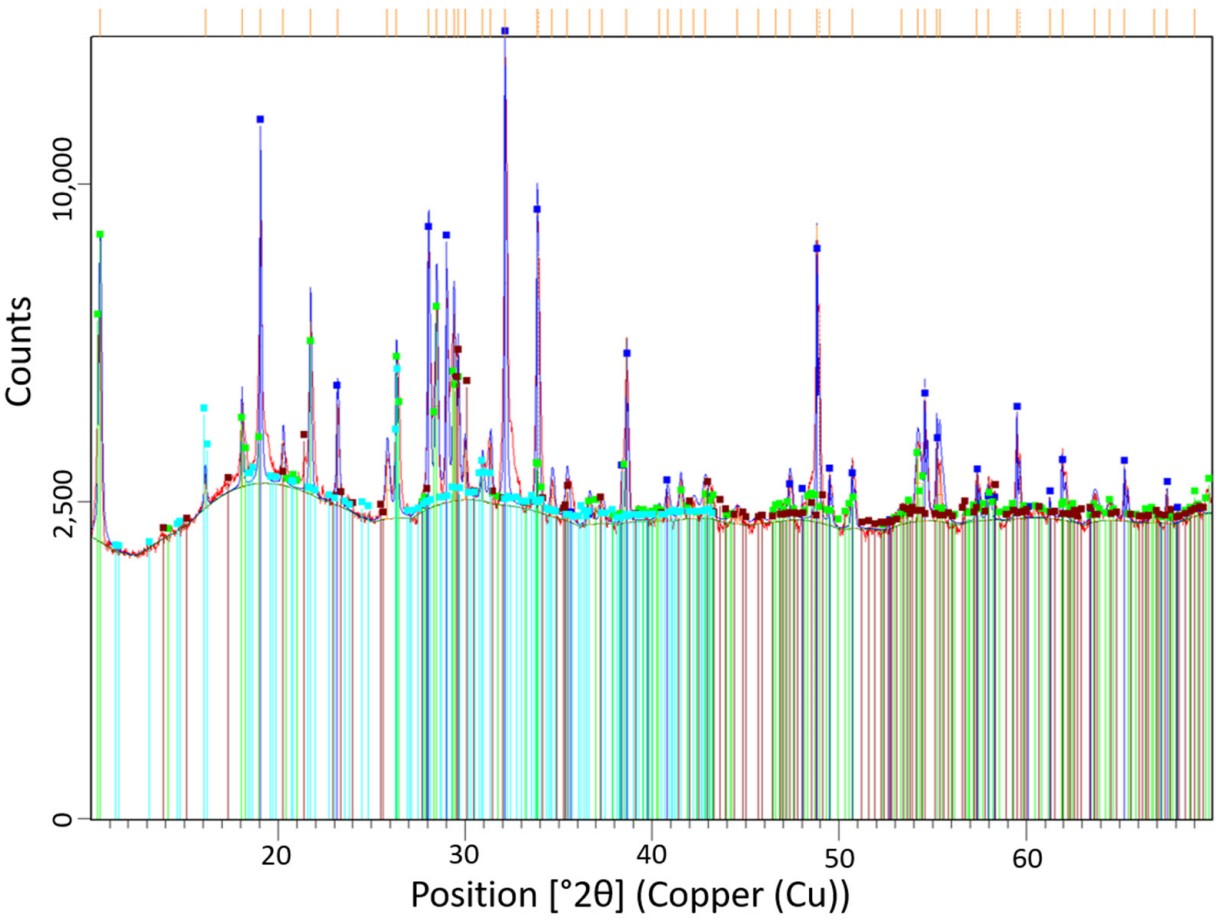

**Figure A1.** Phase identification of compounds in an XRD pattern using HighScore Plus program.

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
