# Peer review of "Implications of the Use of Biodiesel on the Longevity and Operation of Particle Filters"

_lubricants, doi:10.3390/lubricants10100259_

Round 1

Reviewer 1 Report

The paper presents work on the implications of the Use of Biodiesel on the Longevity and Operation of Particle Filters, the paper looks fine and could be published the author should consider the follow:

-Improving the abstract, by defining the main objective of this work.

- Improving the introduction and literature review, by looking at recent work published in this or similar journal.

-Discussions should be improved, author should discuss the current work outcome with previous published papers.

-Conclusions should be specified the main contribution of this work.

Author Response

Thanks for your review. Please see the attached comments.

Reviewer 2 Report

Manuscript: Title: Implications of the Use of Biodiesel on the Longevity and Operation of Particle Filters

 The aim of this manuscript is to add more information to this topic by way of investigating a DFT sample which has been subjected to accelerated aging with Na doped fuel. The results of this study are nothing new, only repeating the research conducted by Lance et al. 2016. The current manuscript can be published in Journal of Lubricant after major revision as mentioned below:

Abstract:

The author state that this study investigates a diesel particulate filter (DPF) which has been aged with 15 a Na-doped biodiesel fuel (B20) and found that the Na-ash accumulated in the DPF. The author just simply repeats the research reported by Lance (2016).

First, the author needs to analyze the amount of Na or K content in biodiesel before using it with an appropriate method.

Second, the author should preparate the samples from used DFT of biodiesel prepared using Na and K catalysts without doping Na or K compound.

The Na or K catalyst used for biodiesel production is only 1% to vegetable oil. Many preparation steps are also carried out to purify biodiesel, such as centrifugal, filtering, washing, and drying to remove residual catalyst sources and methanol.

Suppose the author wants to know the presence of a residual catalyst in biodiesel production; in that case, the authors should characterize the K-deposited-DFT from a biodiesel sample produced using Na+ or K+ catalyst.

In the methodology section:

In the methodology section (Lance et all. 2016), report the accelerated aging of a production exhaust system was conducted on an engine test stand over 1001 h using 20% biodiesel blended into ultra-low sulfur diesel (B20) doped with 14 ppm Na. It is hard to believe the author's opinion because the Na source was not from the sample of the used biodiesel. The Na and K sources should be taken from biodiesel samples, not from samples doped with Na into biodiesel.

 In the Results and Discussion section:

The SEM and XRD images, there are do not show DFT pollution from commercial biodiesel. The authors report the sources of K and Na from the doping process of biodiesel. The more Na and K doping, the greater the results identified by SEM and XRD. Nothing new was found in this report.

Author Response

(The authors gave the same response as above.)

Reviewer 3 Report

This is an interesting and timely paper. I have a few minor comments that may help the authors improve the paper still further.

1. Typo: page 3, line 108. "mush" should be "must"

2. Biodiesel is derived from different types of crops in different regions of the world. The authors do not say what type of crop the biodiesel used in their study is derived from, and it is not clear from the literature review in the paper if the different studies mentioned all use the same type of crop for their biodiesel or if they are different. I think the authors should mention this since if the trial was run in a different geographic region, the results could differ due to the source of the biodiesel. 

3. It is not clear to me if filters from different manufacturers would have the same design or materials, so the authors should comment on whether the specific design or manufacturer of the filters will also impact the results.  

Author Response

(The authors gave the same response as above.)

Round 2

Reviewer 2 Report

Dear Author,

I  already checked the revision sheet and manuscript titled, "
Implications of the use of biodiesel on the longevity and operation of particle filters ".

The manuscript is now ready to publish.

Should you have any inquiries, please do not hesitate to contact me.

 Best Regards,